# Research on the Ethology and Diet of the Stray Dog Population in the Areas Bordering the Municipality of Suceava, Romania

**DOI:** 10.3390/vetsci10030188

**Published:** 2023-03-01

**Authors:** Gabriel Dănilă, Valerian Simioniuc, Mihai Leonard Duduman

**Affiliations:** Faculty of Forestry, “Ștefan cel Mare” University of Suceava, 13 Universității, 720229 Suceava, Romania

**Keywords:** behaviour, food, spread, stray dog

## Abstract

**Simple Summary:**

The population of stray dogs is increasing worldwide, and its management has given rise to ecological, social-economic and ethical problems. In addressing this problem, this study aims to determine the behaviour of stray dogs on the outskirts of a heavily anthropized area, as well as the type/nature of the food consumed. The study highlights the fact that this population of stray dogs thrive in areas that provide them with shelter and food. Additionally, the predatory behaviour of stray dogs is demonstrated, including behaviour similar to that of related wild species (such as the jackal and wolf). The analysis shows the opportunistic nature of the dogs’ feeding, but also their obvious preference for food of animal origin (meat). This kind of regional analysis could be corroborated in order to identify technical and ethical solutions aiming to minimize the negative economic and ecological impacts of stray dogs on human society and the environment.

**Abstract:**

This paper analyses aspects of the ethology and feeding of stray dogs in the areas bordering the city of Suceava and the nearby towns. The study area is located in the hunting grounds (HG), which are managed by the “Ștefan cel Mare” University in Suceava. Between October 2017 and April 2022, the behaviour and the types of food consumed by stray dogs captured in the outskirts of the localities in the study area were analysed. A sample of 183 stray dogs were used for the study, and the analysis established the distribution and density of the dogs in the free-range area compared to the density of the wild animals of hunting interest. The tracks and travel routes of the stray dogs were highlighted. Areas where packs of feral dogs camp were also identified. Observations were made of the individual and social behaviours of the dogs, their gregariousness and the way in which they hunt. The types of food consumed were analysed for each specimen. Through the collected and analysed data, the opportunistic predatory behaviour of the stray dogs was highlighted. Thus, stray dogs revert to the typical wild canid ways of behaving. As for food, our results showed the dogs’ predilection for meat, both wild and domestic. On the other hand, the diet of roaming dogs is much more varied compared to that of wild canids. This is due to the fact that the way in which domestic dogs feed has changed over thousands of years as a result of living alongside humans.

## 1. Introduction

Over time, dogs have become dependent on human presence. Very recent archaeological discoveries in an Egyptian cemetery (the ‘pet cemetery’ at a port that functioned from the mid-1st to mid-2nd century AD, with 585 graves) demonstrated the dependence of domestic dogs on human care. This was established by the pathological lesions and diseases of the dogs, which led to the conclusion that these animals could not have survived unaccompanied [1]. 

Dogs are used to protect property, to reduce human–wildlife conflict, for hunting, as pets or service animals and for entertainment [2]. The worldwide dog population is estimated to be over 500 million [3,4]. According to other sources, the population of dogs on the world map is around 700 million, of which around 75% are classified as stray dogs [5]. 

A study carried out in Latin America identified and measured the following causes of abandonment: aggression (83.6%), illness (38%), behavioural problems (20.9%), change in owners (13.3%), a lack of space (3.8%), old age (3.8%), etc. [6]. However, in certain situations caused by breaks in the relationship of these animals with humans (lack of food, lack of care, overbreeding, etc.), stray dogs have tried to survive in their new conditions, most often by changing their behaviour. In most situations, this behaviour has become transitional, shifting between domestic animal behaviour and wild animal behaviour. The increased sociability of dogs means that their adaptation to the extra-urban habitat is very easy [7,8]. 

According to the legislation in force today, a stray dog is any dog found free in hunting grounds which does not belong to the category of hunting dogs or show a distinct sign attesting to that fact and whose owner cannot be identified at the time [9]. 

Even though dog populations are important in terms of their number, size and distribution, there are few studies on their behaviour and impacts on wildlife and the environment in general [7,10,11,12,13,14,15,16,17,18]. 

When neglected or no longer needed, dogs can become feral or stray. In certain areas, stray dogs have become the most abundant carnivores, significantly disrupting ecosystems [3,19]. Dogs also spread diseases, harass or kill wildlife, and compete with native species. As carriers of pathogens of diseases such as rabies, parvovirus, leptospirosis and coronavirus [20], dogs can cause significant declines in native wildlife populations, some of which are endangered [21]. Our close relationship with dogs and the numerous uses to which we put companion animals and their ubiquitous distribution have resulted in dogs’ and cats’ unwitting participation in the spread of over 60 parasite species, including Giardia, Cryptosporidium, Toxoplasma, most foodborne trematode species, *Diphyllobothrum*, *Echinococcus* spp., *Ancylostoma* and *Toxocara*. Changing human behaviour through education to encourage the proper cooking of food, which may have cultural and social significance, will remain as challenging as controlling stray and feral pet populations, improving hygiene levels, providing safe drinking water and the proper use of sanctuary facilities [22]. 

Although the direct killing of wildlife is their most dangerous effect, many dogs stalk sedentary species, which directly results in increased stress and energetically costly behaviour in native wildlife. The mere presence of dogs discourages the use and occupation of the respective areas by wild animals [23] and may induce negative effects on the reproduction rate of indigenous species such as ungulates [24]. Additionally, dogs act competitively within the predator complex [11,25].

In some cases, the effects of predation by feral dogs may be greater than that of wild predators [26]. Thus, there is a higher rate of predation by stray dogs, especially near human settlements. High densities of stray dogs can also have negative effects on the recovery of small carnivore populations, but this is also true of low densities of feral dogs [27]. 

The danger posed by stray dogs to wildlife is exemplified by the fact that a single stray dog caused a decline in the Kiwi species. The stray dog was discovered when the Kiwi birds were radio-tagged [28]. 

Another study found that faecal samples collected as wolf droppings were misidentified as deriving from stray dogs [29]. The inefficient collection and storage of waste, especially in developing areas, generate a large volume of waste on the streets or in open pits. These wastes (food, animal carcasses, human faces) serve as food for stray dogs. After consumption, stray dogs become vectors for the transmission of various diseases (rabies, leptospirosis, parvovirus, coronavirus) to other animals or even to humans [30,31,32].

Additionally, the socio-economic costs and the traffic safety risks caused by the collision between vehicles and animals have increased in recent decades. The conducted studies showed that of animals the involved, the highest percentage was stray dogs [33].

Unfortunately, most incidents of attack and harassment by stray dogs on wildlife or humans are not reported to a higher regional or national forum for centralization [34].

The relevant Romanian legislation [35] does not currently provide for the centralization of method to control the number of specimens of wild and domestic animals killed by stray dogs. Current regulations only provide for the identification and reporting of the damage caused by large carnivores to both wild and domestic species [36]. Practically speaking, there are no accurate statistics of deaths caused by stray or feral dogs.

Worldwide, studies conducted on stray dog populations, by interviewing owners and other categories of people, have indicated their high impact on wild and domestic animal populations [37]. The objectivity of the respondents is often doubted, as the reports are presented by citizens without specialized training.

Over the years, in Romania, legislation regarding the economy and the protection of game, as well as the status of stray dogs, has been well-established. As stray dogs are considered as pests, they can be removed from free land by technical staff or by hunters, without restrictions or obligation for compensation and without the deed being sanctioned [9,38,39,40]. In fact, one of the mandatory duties of the personnel employed to protect game is to remove stray dogs from the forest or agricultural land.

Animal protection organizations have a completely different perspective and are campaigning for a ban on the killing of stray dogs. These steps taken by non-governmental organizations are often questionable for various reasons. Many times, these interventions are carried out by non-specialists, without solid prior documentation and without providing potential technical solutions for solving the cause of stray dogs. Basically, these protectionist associations call for actions aiming to raise interest groups’ awareness without addressing the problem scientifically and professionally. In this way, various target groups are manipulated, including the mass population, by means of media communication. The shortcomings created by the stray dogs are presented one-sidedly, highlighting only that side that evokes tenderness and pity.

The real danger that stray dogs represent, both for the population and for wild and domestic animals, is deliberately marginalized. The causes of these methods of approaching the problem are multiple and oriented towards pecuniary, image, rating or even political interests [41,42,43,44]. This method of treating the subject can also be born out of naivety or ignorance, but it resonates with the audience. There are also situations where certain associations/organizations use the cover of protecting dogs for illegal activities or even where their employees torment and torture dogs in shelters [45]. On the other hand, there are also bona fide organizations that are genuinely and professionally involved in identifying solutions for stray dogs. Unfortunately, these associations are discrete and less vocal [46]. 

Purpose. This study aims to determine the density of free-roaming dogs, as well as the types of food consumed by them. In this sense, the nature of the food provides us with information regarding the activity of the dogs and the danger they represent for the fauna. With knowledge of the true density of stray and feral dog populations, their configurations and distributions in the wild and how they evolve behaviourally and numerically, intervention techniques can be developed to reduce their numbers in the most precise, organized and efficient way.

The ultimate goal is to reduce the impact of stray dogs on natural ecosystems. For this, analyses are required on the means and the causes of the behavioural transformation of dogs from domestic dogs to wild dogs. It is also necessary to identify a method to quantify the impact that stray dogs have on wild or domestic animals, as is appropriate. The dispersal of stray dogs outside towns, the modification of their behaviour and the sources of food they seek for survival must be analysed.

Finally, all these studies and the cumulative information acquired can be used to determine the identification of ethical and effective solutions so as to minimize the danger posed by the spread of stray dogs in natural or agricultural ecosystems.

## 2. Materials and Methods

The study area was chosen taking into account the fact that urban agglomeration is one of the main factors for the dispersion of stray dogs on the outskirts of, and in places with, important forest areas and agricultural and pasture ecosystems (Figure 1, Table 1).

From an administrative point of view, the area includes several territorial administrative units (UAT), namely the Municipality of Suceava, the City of Salcea and the communes of Mitocul Dragomirnei, Adâncata, Dumbrăveni, Verești, Siminicea and Hănțești.

From the point of view of forestry administration, the forest units in question are managed by Adâncata Forestry and the Pătrăuți Forestry.

The research took place in the territory of two neighbouring hunting grounds (Figure 1), managed by the University of Suceava through the Faculty of Forestry, which are adjacent to the municipality of Suceava [47].

The land uses encountered in the studied territory are as follows: forest—2606 ha (13%), arable—1550 ha (8%) and pastures—12,037 ha (62%) (Table 1).

Of the total studied area of 19,462 ha, 83% represents the productive hunting area, amounting to 16,197 ha, respectively. The area occupied by the forest represents 13% of the productive hunting area and is concentrated in the Mitoc hunting grounds in the north-western part of the analysed region.

From a geographical viewpoint, the analysed area is located in the territory of the Dragomirna Plateau, which belongs to the Eastern European Province, Moldova Plateau Subprovince, Suceava Plateau District [48].

The Dragomirna plateau is positioned on the right side of the Siret River, and the altitude is between 340 m at the Siret River level and 425 m in Culmea Romanoaia.

The method used to collect the data was observation, with observations conducted on route and stationary so as to study the behaviour of stray dogs. Itineraries were predetermined in order to carry out the guarding and observation activities of the field staff, in accordance with their duties. Observations on the itinerary were conducted between October 2017 and May 2022, with a periodicity of about one week, by trained teams consisting of two to three members. The technical staff from the hunting department participated in the actions in the field on the hunting grounds managed by the Faculty of Forestry, including a member of teaching staff (PhD), two engineers and a game warden.

Field trips were conducted in the morning and evening, when the activity of stray dogs is maximal in order to procure food. This circulation occurs simultaneously in all animals, including wild mammal species. The field activity consisted of observing the presence of stray dog specimens and recording information related to their location in time and space, population characteristics, directions of movement, mode of aggregation, etc.

According to the legislation in force, a stray dog is any dog found free in hunting ground which does not belong to the category of hunting dogs or show a distinct sign attesting to this fact and whose owner cannot be identified at the time [39].

The data collected and analysed took into account only stray dogs captured on the productive surface of the hunting grounds outside the village and only specimens identified in the wild, unaccompanied by citizens and without signs of belonging.

It should also be mentioned that the pace of the actions taken to detect stray dogs was relatively constant. In this sense, field activities were carried out almost daily, except during periods of rain, snow or strong wind. This did not affect the rhythmicity of the observations, because during these periods, both stray dogs and wild animals are inactive or minimally active. To avoid possible errors regarding the origins of the stray dogs, no captures were performed in close proximity to the urban perimeter, as there was a possibility that some of the dogs came from citizens’ households, which was a potential source of conflict and would have skewed the subsequent analysis.

To ensure the safety of the capturing activity, a minimum distance of 500 m from the intra-village limit was maintained. This distance was chosen taking into account the provisions of art. 44, para. 16 of the Regulation of 4 June 2008 regarding the authorization, organization and practice of hunting [49]. 

The activity was carried out using off-road vehicles, as well as on foot. Field trips were conducted in the morning and evening, when the activity of stray dogs is maximal in order to procure food.

The identification of dogs in the field was performed with the help of optical devices, mainly binoculars and thermal cameras: Docter 8x58 B/CF binoculars, cameras with built-in GPS systems, Sony DSC-HX60V, Pulsar Axion Key XM30, Garmin GPSMAP 65 GPS, and kits with instrumental equipment for dissection.

The observations on the itinerary were part of the activities of monitoring, guarding and combatting pests within the range of hunting grounds under the management of the university.

Following the combat actions carried out by the technical staff, according to the legislation in force, 183 specimens of stray dogs were captured by shooting (121 specimens on FC 55 Mitoc and 62 specimens on FC 56 Salcea). According to art. 19, paragraph 3 stray cats and feral or stray dogs found in the productive areas of hunting grounds are captured or shot without restrictions and without the obligation to pay compensation, and the act does not need to be sanctioned. Paragraph 4 Cats and dogs that are in the situations provided for in paragraph (3) can be shot by specialist staff involved in the management of the hunting grounds or by hunters upon authorization [9]. The capture of stray dogs does not need to be carried out for research, the capture being a component of the duties of the technical staff who are in charge of guarding the hunting grounds, according to the Law on hunting and the protection of hunting grounds, no. 407/2006. The research team had the opportunity to benefit from this circumstance so as to carry out a study on the feeding behaviour of stray dogs and qualitatively analyse the stomach contents of each specimen. 

After each combat action, the stray dogs were photographed, simultaneously with the automatic recording of the GPS coordinates.

Next, the specimen was dissected in the stomach area, with the qualitative and exact identification of the stomach contents [50]. For each specimen, photographs were taken of the food bolus and its components. The identification of stomach contents was performed visually, taking into account major differences in appearance between domestic and wild animal hair, which is different for each species. The analysis of the stomach contents was performed by the same specialist in all cases.

After making the determinations and completing the field sheets, the analysed specimen was sealed in a plastic bag and transported to the place of storage or incineration, as appropriate.

The analysis and recording of the collected data regarding the locations of the captured specimens were performed in the office by photo-processing with the ArcGIS computer platform (ESRI). Each photo had geographic coordinates (latitude, longitude and altitude) attached, so that the location of the respective specimen was very precise. By processing with the Arctool box module, using the Phototopoint function, a point shapefile was assigned to each photograph, respectively, for each specimen. A characteristic of point-type vector files is the fact that jpg images or different attachments can be attached to the located points, which can later be spatially correlated (Figure 2).

In Figure 2, the access process with the HTML pop.up function of the various points, which represent the identified stray dogs, is exemplified. The image of each specimen contains different identification data, which can be supplemented with subsequent attachments.

The kernel density method calculates the density of certain features/parameters in areas of interest. Possible uses include the determination of densities with a wide variety of points or line features for both urban/rural areas and wildlife habitats. In the present case, this module was used to calculate the density/dispersal pattern of the identified stray dogs. Additionally, the dispersion of the dogs in terms of space and their concentration in certain areas in the total space, including variations by sex, were analysed. 

The primary data analysis and graphical representations necessary to interpret the results were performed using EXCEL (Microsoft Corp., Redmond, Washington, WA, USA). 

## 3. Research Results

The results of the analyses focused on several directions, namely, the dispersion of stray dogs in the analysed territory, aspects regarding their ethology and analyses regarding the foods consumed.

### 3.1. Distribution of Stray Dogs in the Analysed Territory

Through specific processing using the ArcGis computer platform, the spatial distribution of the stray dogs was achieved. Thus, one can observe the distribution in the analysed area and the concentration and dispersion in the field of all the stray dogs captured (Figure 2).

Based on a first analysis of the distribution, one can notice an obvious concentration of stray dogs in areas covered with forest vegetation and in the vicinity of household waste deposits (garbage pits) (Figure 3 and Figure 4).

One can also note the high density of stray dogs, with over 11.3 dogs per 1000 ha of extra-village land (Table 2).

The preference of feral dogs for areas with or near forest and for some areas in agricultural lands or near towns is evident.

The kernel distribution analysis shows a maximum concentration in the Adâncata forest area, located in the north-west of the area (red colour) in the vicinity of the Grigorești-Simincea localities (46.8/1000 ha) (Figure 2, Table 3). Secondly, several smaller nuclei—shown in orange and orange-green colour in the image—can be seen near the town of Plopeni (Figure 2) in the vicinity of the garbage pits and the agricultural lands in the north of the town, where small game are present.

In the territory devoid of forest vegetation in the Salcea–Dumbrăveni–Suceava river meadow, the dogs showed a relatively uniform distribution and a much lower density (4.5/1000 ha).

Regarding the spatial distribution by gender (Table 2, Figure 4), an uneven distribution of males and females can be noted, even if the sex ratio is relatively balanced.

Overall, the difference is relatively small, with males predominating at 53%, compared to 47% females (Figure 4 and Figure 5). However, the fact that males predominate indicates a potential increase in the intensity of natural selection, which can become amplified over time.

If we analyse the spatial distribution of the stray dogs by sex, observing Figure 5, a higher density of males can be found in areas without forest vegetation (about 35 males and 26 females). A balanced, equal number (about 61 males and 61 females) is observed in the areas with forest units.

### 3.2. Aspects Regarding the Ethology of Stray Dogs in the Analysed Area

By analysing the modes of the spread of stray dogs based on multiple observations, which took place within approximately the same timeframe, certain areas frequented by the stray dogs were identified. These areas are shown in Figure 4 and Figure 5. Through both the all-terrain vehicles and the outline of the patrolling areas, the movement routes of the stray dogs were identified. Two large categories of stray dogs could be observed, which followed these trails.

The first category is composed of specimens that moved from towns to agricultural areas and to the forest. This first category includes both urban stray dogs with owners disinterested in the fate of the dogs but also semi-feral stray dogs that live on the outskirts of towns and have no owners. Stray/semi-feral dogs from the inner city and in rural areas occupy abandoned houses left in disrepair for various reasons.

The second category includes completely feral roaming dogs that live only in the countryside.

Depending on the topography of the terrain, the dogs in the first category almost strictly kept to certain routes, most often where the terrain benefits them, where the trails follow streams, creeks or ravines of various categories, or areas with shelter belts, fences or tree lines, similar to wild canids (foxes, jackals). Such routes were observed in the “slag pit” area near the Suceava Municipality, along the left bank of the Suceava River, leading to the agricultural lands belonging to the towns of Salcea, Plopeni and Prelipca. Other routes were observed from Salcea to the Fetești forest, from the Burdujeni neighbourhood to the Adâncata forest, from the Ițcani neighbourhood to the Rusciori forest, from Mitocași to the Romanoia forest and from Hânțești–Adâncata to the Siretului river and to the Pleșa forest (Figure 5). These main routes are supplemented by shorter routes, which are less frequented but compose the network of travel paths in the analysed territory.

The nuclei of the completely feral stray dogs was in the Dumbrava–Rusciori forest, situated on the ruins of a former ammunition depot, an area later taken over by the former AVICOLA Suceava. In this area, the dogs use all the bumps and the former cellars of the warehouse, as well as the technical channels of the former water, sewage and fire installations, etc. The dogs in this area have a clear wild behaviour; they cannot tolerate proximity to people, and if they are in a pack, they may even attack people who are passing by.

Another important nucleus occupied by the former CET Suceava (Electro-Thermal Power Plant) comprised the tunnels positioned along the railway in the Văratec–Salcea area. Two other areas that harbour feral stray dogs are in the Plopeni area, along the Mereni stream. This stream is ravined with very steep banks (slopes of over 45%), on which a very dense shrubby and sub-shrubby vegetation has developed, consisting of hawthorn, species of willow, rosehip, elder, willow, sea buckthorn, dovecot, etc. There are also exclusively wild stray dogs in the area of the sewage settling basins in the former pig breeding complex, the former ISCIP Verești (Figure 6). 

In other areas, especially in the central and southern parts of the analysed area, where the terrain is almost flat and there are no conditions for trails on which dogs can move unnoticed, tracks can be observed in the agricultural land or pastures.

These paths/trails were easily identified during the winter, when there was a layer of snow. Later, these travel paths were also checked in other seasons, after rainy periods, when the tracks were visible, and the age, as well as the number and size, of the specimens could be approximated. Additionally, during the winter, it was found that the movements of the stray dogs took place on the tracks of other animals and on the tracks of carts or vehicles, which was obviously due to the need to conserve energy. In the winter, the travel paths became almost flat due to the high frequency of travel.

Feeding behaviour of stray dogs. Circadian movements are mainly carried out in the search for food. Stray dogs have opportunistic feeding behaviour. First of all, this behaviour focuses on all the points where UATs or citizens legally or illegally store various types of household waste. These warehouses are located at variable distances of 2–3 km from the town limits.

In the activity of searching for food in large territories, individuals or groups form skills for movement on certain routes that ensure minimum energy consumption and maximum protection.

Both semi-feral and feral dogs have access to these food sources. The main deposits are on the banks of the Suceava River, the CET area, the Verești ballast area, the Dumbrăveni slaughterhouse area, the Siminicea ballast yard, the Țigăncii ravine of Hănțești, the Odăi Adâncata clearing and the exit from Fetești (Figure 7). In addition to these points, much debris can be found along the watercourses and in the forest, where citizens illegally deposit garbage bags and household waste.

Apart from this somewhat easy feeding pattern induced by the adaptability of the stray dogs, the hunting instinct is also manifested for feeding. It is clear that one cannot compare the nutritional quality of household scraps with the properties of meat. For this reason, feral dogs mainly hunt wild but also domestic animals (Figure 8, Figure 9 and Figure 10). Their menu includes wild animals, including small rodents such as water shrew (*Neomys fodiens*), vole (*Microtus arvalis*), ground squirrel (*Spermophilus citellus*), hamster (*Cricetus cricetus*) and hares, as well as young and adult roebuck and young wild boar.

It was observed that the hunting of the wild specimens was carried out, as is typical of canids, by stalking and flanking the prey (Figure 11). Most often, they preyed on juveniles and animals injured for various reasons, or those with certain ailments. For example, the area being humanized, many deer or juvenile wild boars become injured in wire fences or are hit by agricultural machinery or vehicles on the roads.

These specimens clearly became victims of wild dogs, since there is no other predator in the area. Additionally, hare predation is based on hares’ low endurance over long distances compared to the physical endurance of dogs. However, most of the time, the dogs abandon the chase. The same cannot be said of young animals, the predation having the greatest impact on young hares.

The meat of domestic animals mainly includes sheep, both those discarded by the owners as a result of death due to disease and isolated or stray animals, which are hunted in the wild or even left in unguarded pens (Figure 10).

Technical staff were on site during a stray dog attack that resulted in over 30 sheep being killed or injured by a pack of 7 stray dogs. It was evidently the same mode of behaviour as that of the wolf, which instinctively kills more than it needs to consume. Unfortunately, the owner did not want to make a complaint, nor did he want to countersign the report prepared by the technical staff. Similar incidents also took place in Mitocași, with 5 dogs killing 13 sheep. In this case, the shepherd was firmly convinced that the attack was caused by jackals, being “white with black spots” (sic!). Damage was also caused to three juvenile bulls in the area of the slaughterhouse in the town of Dumbrăveni. Additionally, more examples could be mentioned.

Over time, we were informed many times by various citizens, including sheep owners, that they have suffered significant damage, ranging from 1–2 sheep to over 30 sheep in one night. Unfortunately, these numbers have not been quantified in official reports or minutes at town halls or by the police department, because in most cases, citizens avoid filing or refuse to file a complaint.

The gregarious (social), reproductive and territorial behaviours of stray dogs are manifested through the social relations that take place at the level of the respective group. Similar to other gregarious canid species (jackals, wolves), semi-feral and feral dogs communicate in a complex way through visual, auditory or olfactory signals. As a rule, it has been observed that the hierarchy in the group/pack is dictated by size and age, respectively, and by the experience of the component specimens among whom subordinate relations are established.

Basically, in the same group, short conflicts take place as a result of which the weakest dog recognises the dominance of the others but shows its authority over others who are weaker than it. This method of communication, which is typical of canids, takes the form of barking, growling, a certain positioning of the tail and ears or baring of the teeth. Ears laid back and the baring of canines indicate hostility. This sum of attitudes provide cues to other group members about aggression or submission. As a rule, direct conflicts are relatively rare between members of the same group and occur only when individuals are close in size.

More intense conflicts arise when an individual from outside the group appears, and in this case, the phenomenon of tolerance can appear, with the intruder adopting a defensive attitude from the beginning and approaching the other dogs very cautiously, presenting itself with its tail between its legs, with its head down in an obvious attitude of submission. In other cases, upon a signal given by a member of the pack, the whole group rushes upon the newcomer and puts it to flight. In such situations, harsh conflicts can take place, from which the intruder can incur serious injuries. Another type of attitude has also been observed, when the roaming specimen is a female and in oestrus. Thus, the female can arouse the interest of the males, and the behaviours change. The manifestations of friendship are obvious; the dogs wag their tails left and right, do not reveal their teeth, and have a visibly playful behaviour.

Sexual behaviour is primarily related to the manifestations of partners during the reproduction (mating) period, though not exclusively, as mating has been observed to be triggered by the time when females enter oestrus. Most “wedding processions” and the most common behaviours are observed from the end of February to March but also in autumn. However, mating in stray dogs can occur year-round due to the domestication of the species.

Due to the fact that they have access to food and shelter year-round, the breeding period is no longer restrictive, limited to a certain interval in a certain season. Practically, there is no longer a distinct mating period as there is in wild species. The timing of the onset of oestrus in females is dictated by the cessation of lactation of the last generation. If, for one reason or another, all the cubs die, the female can mate again. This phenomenon can also be observed in cats and in some species bred in captivity, amongst which the mating period has undergone changes compared to wild species.

In the case of stray dogs, territorialism is very obvious, intruders being repelled up to a certain distance, usually 150–300 m, more often in open spaces and less often in the forest. The phenomenon of territorialism also manifests as directed towards people, and in this case, feral and semi-feral dogs become very aggressive, and most of the time, the pack waits for a leader’s signal in order to initiate an attack.

During the course of the observations, violent tendencies were observed very rarely within the larger or smaller groups. However, the unity of the packs was noted, whether it was expressed against some foreign stray dogs or against citizens on foot, on bicycles or in cars.

Stray dogs have predominantly diurnal behaviour, with a maximum intensity of activities in the morning and evening. However, nocturnal movements were also identified, albeit to a lesser extent. During the day, dogs retreat to more sheltered areas, under the protection of vegetation and under the banks of ravines or streams. In winter, they seek shelter “under the wind” on portions of land without snow, but on sunny days, they resume their activities at noon. In summer, they seek shady areas around forest edges, isolated trees or tall vegetation (Figure 12).

In the periods preceding or following feeding, stray dogs tend to their fur through specific behavioural manifestations of hygiene, including licking or rolling in the snow, and in general, the behaviour of feral dogs is common to all canids, with certain peculiarities exposed previously.

### 3.3. Food for Stray Dogs

The analysis of the stomach contents was performed rigorously and had a qualitative character. According to the nature of the identified ingestions, the number of dogs was determined for each category of food (Table 4, Figure 13 and Figure 14).

For an easier demonstration of the types of food, it can be noted that 103 dogs (56.3%) ingested meat, either from wild animals or from domestic animals (Table 4, Figure 8). One can note the relatively high number of specimens, with 37 dogs (20%) having empty stomachs, which proves that they were seeking food.

Additionally, in total, it was observed that 71 dogs consumed only food of animal origin, and 40 of them only consumed game meat.

On the other hand, wild animal meat was identified in 53 dogs, respectively, amounting to 29%, and domestic animal meat was found in 54 specimens, respectively, amounting to 30% of all the captured dogs.

Only 25.7% of the specimens (47 specimens) showed other different combinations of stomach contents: meat, household materials, vegetables and plastics. It should be noted that the foods of animal origin also appeared in other specimens, in the combinations mentioned above. In this sense, specimens in which several types of ingested foods were identified had animal food/meat in their stomach contents. The number of stray dogs in which the same types of ingested components were found in different combinations is presented in Table 4 based on the total surface area.

Food of domestic origin was found in 18% of the specimens. It is interesting to note that a high percentage of 25% of wild dogs feed on plant-based foods.

A small percentage presented traces or elements of plastic, ingested along with food scraps from households and food thrown into pits/landfills (about 5.46%).

The spatial distribution of the 183 stray dogs by the types of food identified was processed with the ArcGis computer platform as the kernel distribution (Figure 15).

Next, an analysis of the numerical distribution of the dogs by type of food and by season was carried out on the total analysed area. As we can see from Table 5 and Figure 16, domestic-type meat was present in the spring in 12 dogs, i.e., 24% of the total of those who ate in this season, and in the autumn, it was present in 9 dogs, i.e., 18.8% of the total of those who ate in that season.

In the case of domestic animals (in brown colour), a lower percentage was noted in winter and a higher one was noted in spring.

Next, the calculation of the percentage of stray dogs was performed depending on the size of meat identified, either from a domestic animal or from a wild animal. A small size is considered to include specimens weighing less than 9 kg, a medium size is between 9 and 23 kg, and a large size is more than 23 kg [51]. 

From Table 6, one can firstly observe a disproportion of the number of stray dogs captured in different seasons. Thus, most feral dogs were identified in winter (30%).

From a percentage point of view (Table 6, Figure 15), it can be noted that game (in red colour) are part of the diet of stray dogs at a high percentage in winter, summer and spring.

Due to the large number of medium-sized dogs, the percentage of dogs who consumed meat in this category is obviously the highest. However, if we compare meat-eating dogs of the same size, it can be observed that the large dogs have an indisputable preference for this type of food (Table 6).

Overall, we can observe, as expected, the predilection of stray dogs for meat, regardless of its origin, a fact demonstrated by the share of this type of food, amounting to over 56%. Game meat was identified in 29% of the specimens and in 22% of the total dogs feeding only on game meat.

## 4. Discussion

Even though the study area is heavily anthropized, a high diversity of plant and fauna species, as is typical for the relief and climate of the area, can be found in the outskirts of the towns. Thus, the main forest woody species are common beech (*Fagus sylvatica*), sessile oak (*Quercus petraea*), common oak (*Quercus robur*), wild cherry (*Prunus avium*), sycamore (*Acer pseudoplatanus*), elm (*Ulmus*) and ash (*Fraxinus*), and among the species of grassy plants, we can mention lily of the valley (*Convallaria majalis*), shepherd’s sedge (*Capsella bursa-pastoris*), yellow bedstraw (*Galium verum*), bellflowers (*Campanula persicifolia*), two-leaf squill (*Scilla bifolia*), plantain (*Plantago*), chamomile (*Matricaria recutita*), St John’s-wort ((*Hypericum humifusum*), white dead-nettle (*Lamium album*), marsh mallow (*Althaea officinalis*), common yarrow (*Achillea millefolium*), common horsetail (*Equisetum arvense*), chicory (*Cichorium intybus*), colts foot (*Tussilago farfara*), wormwood (*Artemisia absinthium*), common reed (*Phragmites australis*), bulrush (*Typha*), lesser pond-sedge (*Carex acutiformis*), water pepper (*Persicaria hydropiper*), etc. [52]. 

The native fauna species existing in the analysed area are red deer (*Cervus elaphus*), roe deer (*Capreolus capreolus*), fallow deer (*Dama dama*), wild boar (*Sus scrofa*), hare (*Lepus europaeus*), badger (*Meles meles*), otter (*Lutra lutra*), wildcat (*Felis silvestris*), pine marten (*Martes martes*), beech marten (*Martes foina*), weasel (*Mustela nivalis*) and stoat (*Mustela erminea*), and among the birds, grey partridge (*Perdix perdix*), wild duck (*Anas platyrhynchos*), grey heron (*Ardea cinerea*), great cormorant (*Phalacrocorax carbo*), common swan (*Cygnus cygnus*), species of thrushes, species of the Falconidae and Strigidae family, etc. [47]. The existence of predator species, including mammals and birds, proves that the food chains are complete, which, in turn, proves that the ecosystems are relatively “healthy”.

The area of interest is heavily anthropized, but the forest ecosystems, hayfields, pastures and agricultural lands offer good conditions for the existence and development of a wide range of herbaceous and woody plants. The primary plant producers support and provide relatively good conditions for mammal and bird populations that are well-represented, both numerically and in density. The agricultural land, at certain times of the year, also functions as areas that provide food, shelter and peace for wild animals and birds that have adapted to this type of ecosystem.

Stray dogs are dispersed in the open lands, forest ecosystems, agricultural land, hayfields and pastures in higher or lower densities. Clearly, forest units provide them with more opportunities for feeding, shelter and breeding compared to agricultural land/pastures. The wooded areas have a higher concentration of wildlife, which attracts all types of predator. On the other hand, household waste deposits have a decisive influence on the spatial distribution of the stray dogs. These concentrations of debris represent centres with concentrations of stray dogs, to and from which daily or seasonal movement routes begin. On the other hand, in the studied territory, the ungulate species (deer, wild boar) have no natural predator; hence, this ecological niche can be occupied by wild dogs [53]. Through their natural abilities, including that of hunting in packs, they exert predation pressure, especially on deer and hares (*Capreolus capreolus* L., Figure 17), but this can also affect the young contingents of the wild boar species (*Sus scrofa* L.).

It is worth mentioning that the deer population in the study area has a density of 7.4 deer per 1000 ha, with a wild boar density of 4.2 specimens per 1000 ha and hare density of 7.5 specimens per 1000 ha. The average density of stray dogs in the entire analysed area is 11.3 dogs per 1000 ha (total 183 exemplar). This value is very high if we make a comparison with the density of wild species [54]. In the territory devoid of forest vegetation in the Salcea–Dumbrăveni–Suceava river meadow, the dogs have a relatively uniform distribution and a much lower density, the predation/feeding pressure being concentrated on smaller animal species, such as hares and various rodents.

As with any wild animal, the ratio between the sexes provides us with information about the potential for population growth (the case of more females F>) or the increase in the intensity of natural selection (the case of more males M>) [54]. In the area studied, we can note, on average, a sex ratio of about 1:1, which is the same as the natural sex ratio in canids of similar size (jackal, fox). The number of females is much lower in the lands without forest vegetation compared to the forest stands. This discrepancy is caused by the fact that females need sheltered areas to raise their young.

Stray dogs have a relatively wide activity range of several kilometres due to their food resources, and the movement from the shelter area to the feeding or hunting places is carried out on well-marked routes that are thus chosen for the conservation of individual energy and for protection.

In the case of stray dogs, it is difficult to analyse feeding behaviour, gregarious behaviour, reproductive behaviour and territorialism separately, as they are intertwined. During movements of larger or smaller groups, interactions between individuals in the pack, mating displays and feeding take place.

As shown above, stray dogs are opportunistic and consume a wide variety of food types, a behaviour induced by domestication. However, as with all canids, there is an undisputed preference for food of animal origin. On the other hand, 25% of the wild dog specimens fed on vegetable food. This can be attributed to the fact that many canid species consume fruits or plant elements, including blueberries, apples, grapes and even potatoes [55]. Dingo dogs and pariah dogs in India consume small animal species and various residues from garbage dumps as part of their diet. Dogs who are semi-feral and accompany African nomads do not consume anything from their masters but consume carcasses derived from livestock, worms, beetles, grasshoppers, fresh horse dung, etc. Tierra del Fuego’s dogs prefer seashells, which are picked off the rocks with a kick of their paws. Fish is the basic food of dogs in arctic regions, which they catch themselves [55,56]. 

It should be noted that none of the analysed stomach contents contained any elements that attested to the fact that birds were part of the dogs’ diet. This is due to the fact that there are no or few ground-nesting birds (pheasants, quail) in the area.

The mode of the spatial distribution of the food types does not follow a particular rule, and it is random. The distances between the outskirts of the towns and the areas where fauna species live are relatively short, on the order of kilometres, and the dogs can travel long distances in a short time. For this reason, various food mixtures can be identified throughout the analysed area.

It should be noted that most of the stray dogs were captured in winter, including 61 specimens. The main reason for this is the good visibility resulting from vegetative rest, specifically the lack of grassy vegetation and the foliage of trees and shrubs. Additionally, the contrast of the darker colour of the dog specimens on the white snow renders them more visible and easier to follow. In addition, during the winter, mammals need a higher amount of energy, which leads to an increased need for food in order to maintain body temperature, and this fact can accentuate the predation phenomenon in the cold period. The numbers of dogs identified in autumn and spring are equal (48, 49 individuals) but lower than in winter, because in these seasons, there is still vegetation but no snow. In the summer, the vegetation is highly developed on agricultural land and in the forest, as well as in ravines, bends, tributaries, streams, etc., which makes the dog specimens more difficult to track.

The analysis showed higher amounts of game meat consumed in winter and summer, and the reason for this lies in the fact that in the cold season, the phenomenon of predation is easier for the dogs as a result of the gregariousness of various game species (cervids, wild boar) and the use of their tracks/paths for movement. In the summer, predation is concentrated on the young contingents of game, which are much more vulnerable. In this season, herds of specimens of the wild species are at their maximum levels. In contrast, larger quantities of domestic animal meat were found in the spring, and very little was found in the winter. During the winter, domestic animals camp in households and no longer cross the lands bordering the towns. On the other hand, in the spring, there is a high percentage of mortality among domestic animals, which can be found on various paths in the forests, creeks, streams, etc., most often having been transported by the owners themselves, where they are found by dogs and eaten.

Dogs are opportunistic animals, and even though they are considered carnivores, their diet is relatively broad, as demonstrated by the variety of types and mixtures of food bolus identified. 

This research is limited by the size of the area on which it was carried out, but it can be generalised for similar social, economic, legislative and ecosystem conditions.

The economic damage caused by stray dogs is difficult to quantify, as it is related to negative influences on the fauna, the medical expenses necessary to prevent or treat certain diseases and the expenses required for catching and isolating dogs, as well as the direct damage caused by the killing of domestic animals.

Legislative loopholes, as well as inconsistencies between the existing laws that refer to stray dogs, have split the population into two groups with different opinions. Realistic public debates and discussions presented coherently by specialists, as well as some regional local measures, could have the effect of reducing the number of stray dogs. A first solution that would help to reduce the number of stray dogs is to establish sufficiently large and properly financed shelters.

## 5. Conclusions

The domestic dog exhibits a remarkable level of phenotypic diversity, and it is the most morphologically variable land mammal [57]. It is estimated that the population of dogs on the world map is about 700 million, of which about 75% are classified as stray dogs [5].

In the analysed area, the forest ecosystems, meadows, pastures and agricultural lands offer good conditions for the existence and development of mammal and bird populations that are well-represented both numerically and in terms of density. In this heavily anthropized area, there are no large predators (wolf, lynx). Hence, stray dogs are a danger to mammals, birds and domestic animals but also to humans.

The methodology of this study was based on the itinerary and stationary observation method, as well as the capture of stray dogs. A total of 183 specimens of stray dogs were captured, and the stomach contents were qualitatively identified (the nature of the elements of the food bolus).

One can notice a concentration of stray dogs in areas covered with forest vegetation, as well as the vicinity of household waste deposits (garbage pits), being about 10 times higher than that observed in the lands without forest vegetation. Additionally, a higher average density of stray dogs than the average densities of the wild species of interest was observed. This fact is concerning for native fauna and could cause imbalances in the structure of food chains.

Stray dogs can roam and occupy a vacant ecological niche in the food chain. Observations of the behaviour of feral dogs led us to highlight two categories of stray dogs, the first consisting of specimens that travel locally, in the areas outside the village, and the second including completely feral stray dogs from outside the village. The first category consists of urban stray dogs and semi-feral stray dogs living on the outskirts of towns.

The behaviour of stray and feral dogs is identical to that of other gregarious canids, such as the wolf and the jackal. It was observed that the predation on wild specimens by feral dogs is carried out by hunting methods typical of canids, such as stalking and flanking prey, which included mammals of various sizes, ranging from small rodents to young and adult deer, but also domestic animals. Like all gregarious canids, semi-feral and feral dogs communicate in a complex way through visual, auditory or olfactory signals, and group/pack hierarchy is dictated by size and age.

The foods eaten by roaming dogs demonstrate the temperament of an opportunistic carnivore. The foods of stray dogs include several types, such as domestic animal meat, wild animal meat, household organic matter, vegetable matter, plastic and various combinations thereof. Over 56% of dogs consumed meat alone or in various combinations, either from wild or domestic animals, while over 38% consumed meat alone, 29% consumed wild animal meat and over 21% consumed wild animal meat alone.

In countries where legal provisions regarding stray dogs and waste management are clearer, more restrictive and more rigorously enforced, the number of stray dogs is lower and, as a result, their behaviour and feeding may be different. On the other hand, the behavioural and feeding traits specific to canids will manifest themselves whenever the living environment allows them to do so.

## Figures and Tables

**Figure 1 vetsci-10-00188-f001:**
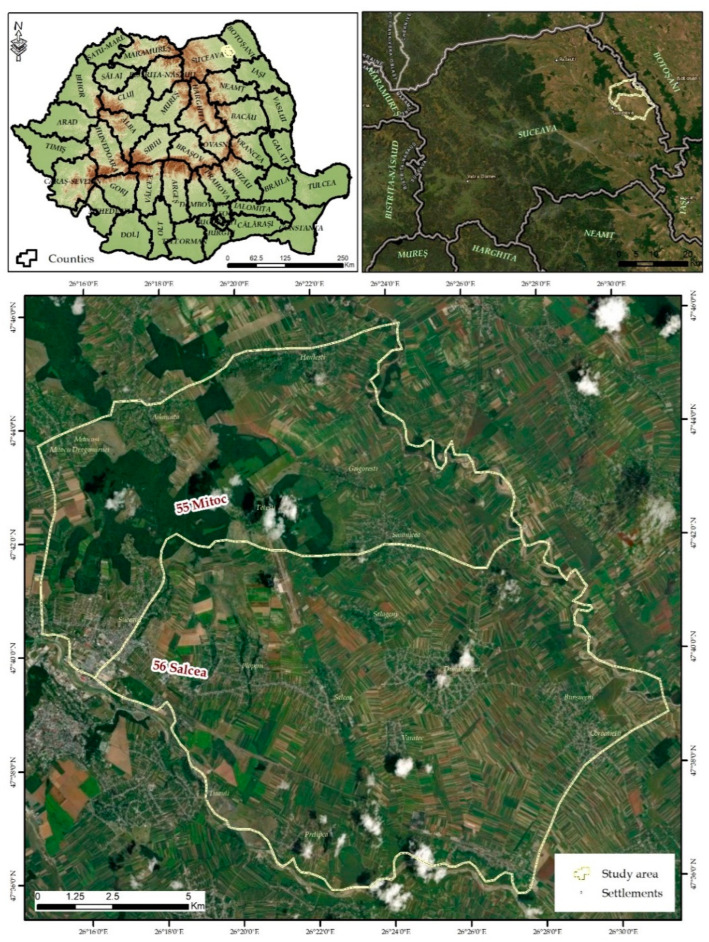
Geographical location of the study area.

**Figure 2 vetsci-10-00188-f002:**
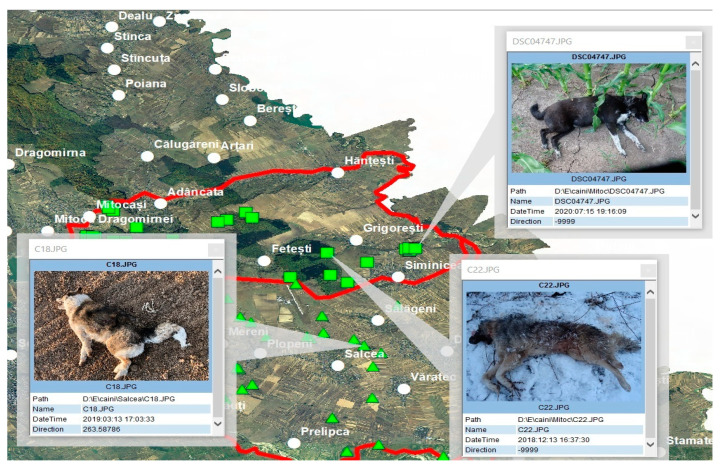
Accessing files through the HTML pop.up function (photos overlay—geotagged photos to points).

**Figure 3 vetsci-10-00188-f003:**
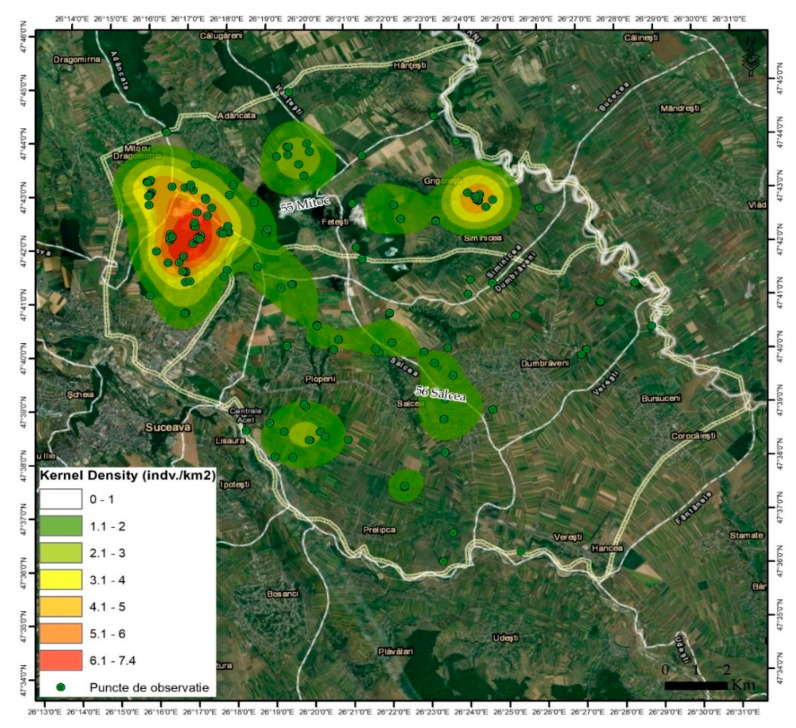
Spatial distribution of stray dogs in the total research area.

**Figure 4 vetsci-10-00188-f004:**
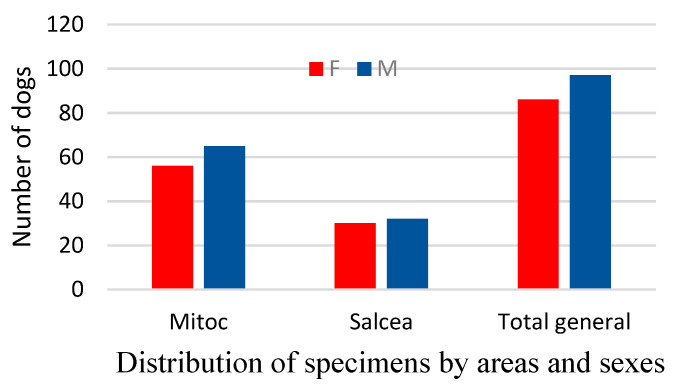
Distribution of stray dogs by area and sex.

**Figure 5 vetsci-10-00188-f005:**
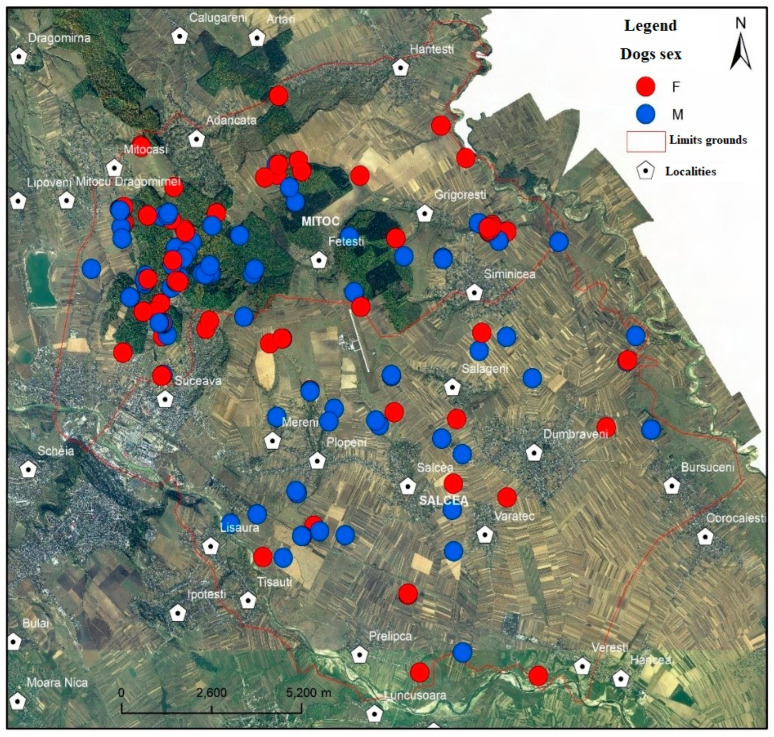
Spatial distribution of stray dogs by sex in the total research area.

**Figure 6 vetsci-10-00188-f006:**
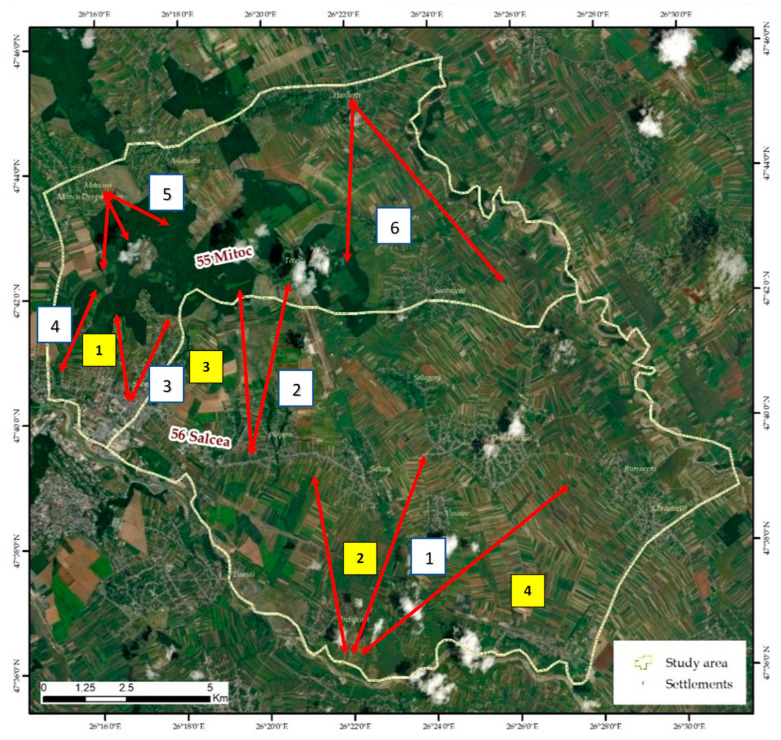
Positioning of routes and nuclei of wild dogs: 
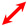
 Routes: 1. “Slag pit” to the agricultural lands of Salcea, Plopeni and Prelipca. 2. Salcea to the Fetești forest. 3. Burdujeni to the Adâncata forest. 4. Itcani to Rusciori forest. 5. Mitocas to the Romanoia forest. 6. Hânțești–Adancata to the Siret River, Pleşa forest. Nuclei: 
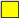
 1. Dumbrava–Rusciori. 2. CET Suceava tunnels. 3. Mereni brook. 4. ISCIP Verești area.

**Figure 7 vetsci-10-00188-f007:**
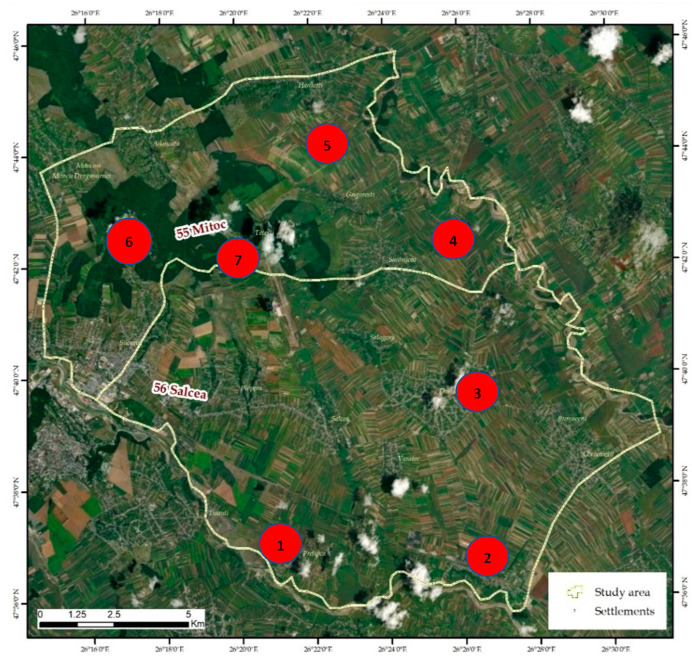
Positioning of areas with deposits of various types of household waste: 1. CET Suceava area; 2. Verești ballast area; 3. Dumbrăveni slaughterhouse area; 4. Siminicea ballast area; 5. Brooks of Țigăncii Hănțești; 6. Odăi Adâncata; 7. Exit of Fetești village.

**Figure 8 vetsci-10-00188-f008:**
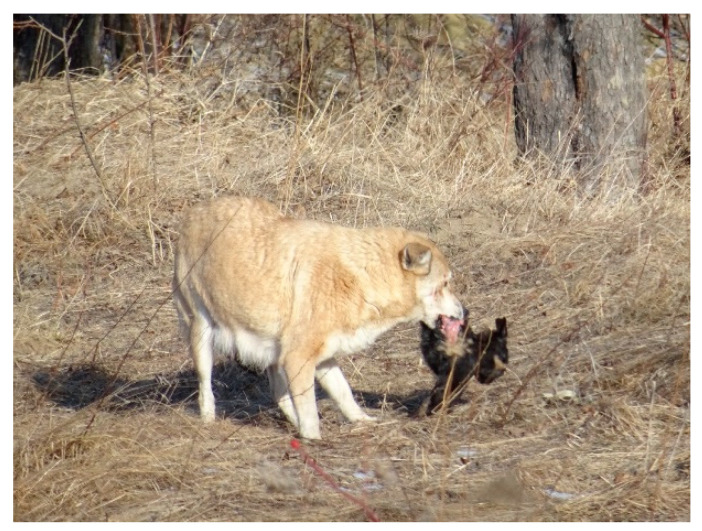
Dog that eats leftovers from domestic or game meat.

**Figure 9 vetsci-10-00188-f009:**
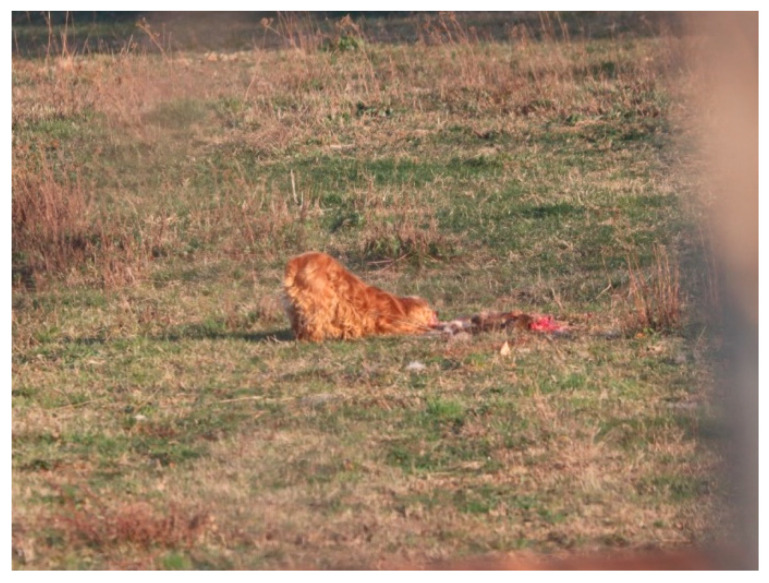
Dog that consumes a roebuck (Odai glade).

**Figure 10 vetsci-10-00188-f010:**
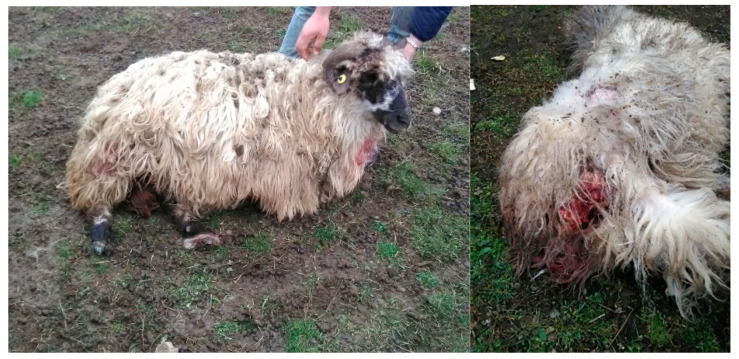
Sheep attacked by stray dogs; injuries caused; place of attack (Mitocasi).

**Figure 11 vetsci-10-00188-f011:**
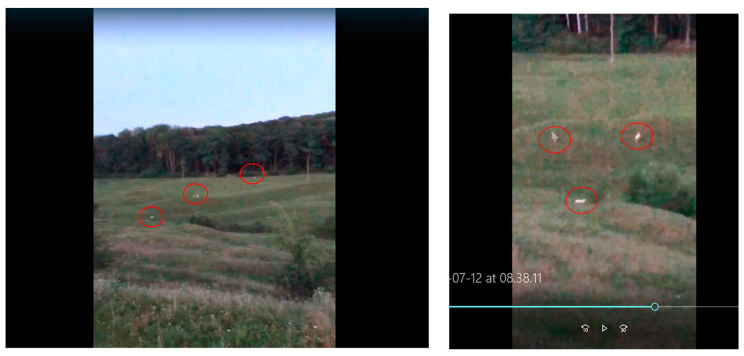
Sheet of wild dogs “in fans” on the hunt.

**Figure 12 vetsci-10-00188-f012:**
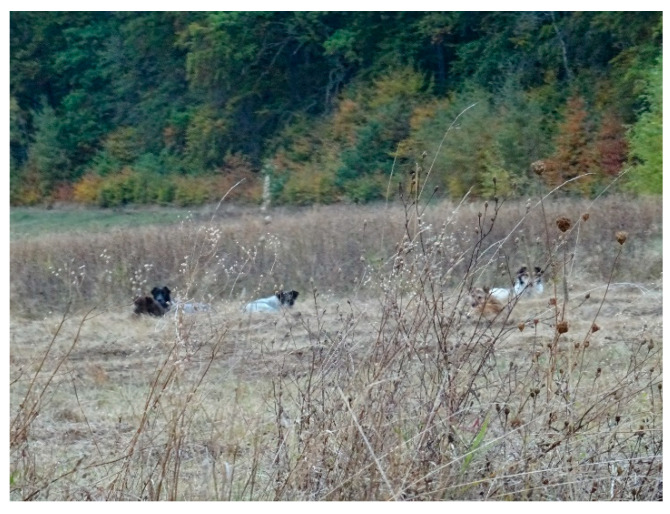
Wild dogs at rest.

**Figure 13 vetsci-10-00188-f013:**
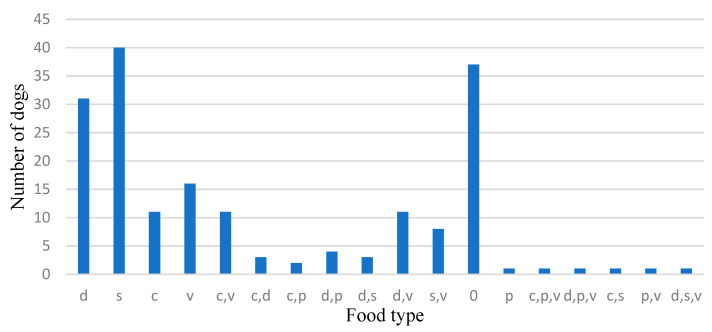
Numerical distribution of dogs according to stomach contents over the total area analysed.

**Figure 14 vetsci-10-00188-f014:**
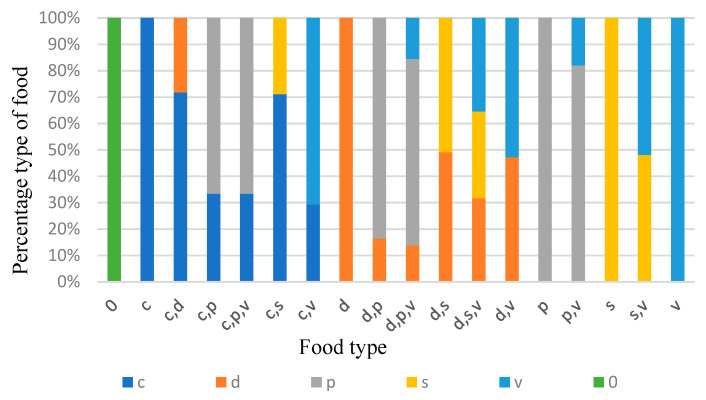
Percentage distribution of food preferences of dogs over the total area analysed.

**Figure 15 vetsci-10-00188-f015:**
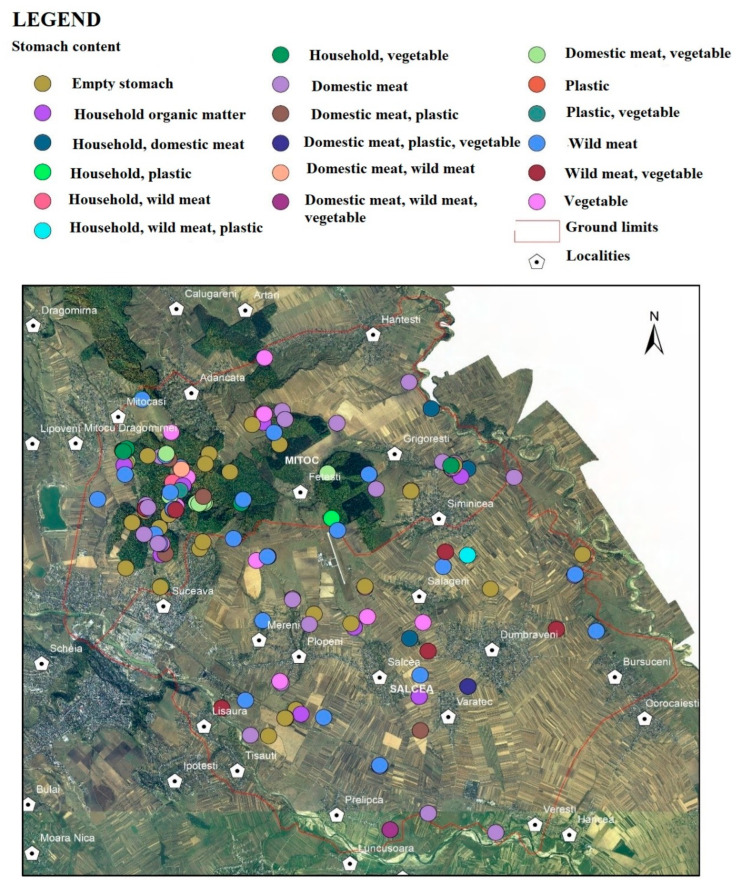
Spatial distribution of food types.

**Figure 16 vetsci-10-00188-f016:**
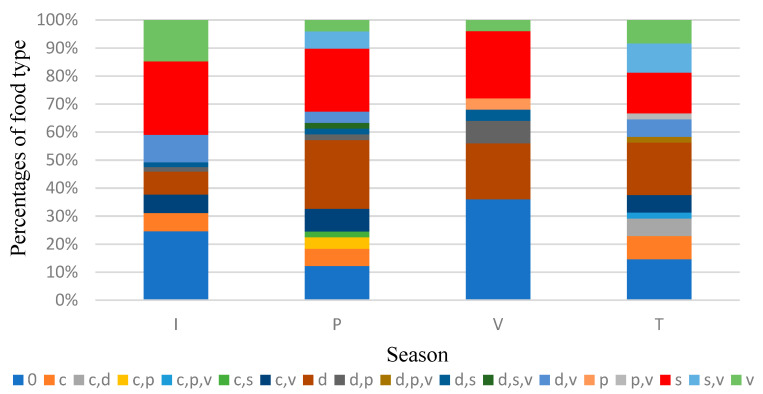
Percentage distribution of preferential foods by season over the total area analysed.

**Figure 17 vetsci-10-00188-f017:**
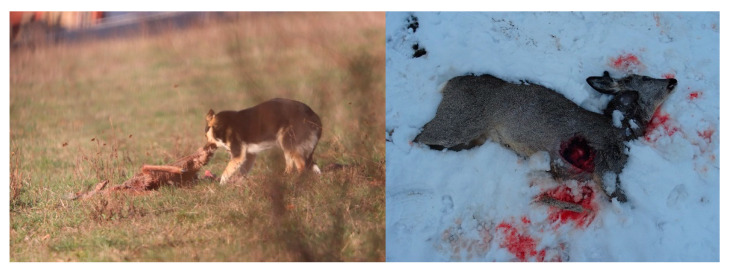
Roebuck specimens preyed on/eaten by stray dogs.

**Table 1 vetsci-10-00188-t001:** Area of hunting grounds by category of use and total study area.

No. Crt.	Hunting Ground	M.U.	Productive Hunting for:						Unproductive Hunting Water Surface	Total Forest
			Marsh Game	The Remainder of the Game Species						
			Water Surface	Forest	Arable	Pastures	Timberline	Total		
1	2	3	4	5	6	7	8	9	10	11
1.	55 Mitoc	ha	-	2550	350	2750	-	5650	1510	7160
		%	-	36	5	38	-	79	21	100
2.	56 Salcea	ha	50	56	1200	9287	-	10,547	1705	12,302
		%	-	-	10	76	-	86	14	100
	Total	ha	50	2606	1550	12,037		16,197	3215	19,462
		%	-	13	8	62	-	83	17	100

**Table 2 vetsci-10-00188-t002:** Density of stray dogs by sex on the level of the study area.

Sex	F	M	Total Stray Dogs	Area (ha)	The Density at 1000 ha		
					F	M	Total
Grand total	86	97	183	16197	5.3	6.0	11.3

**Table 3 vetsci-10-00188-t003:** Density of stray dogs by sex on the level of the study area.

Use	Forest	Agricultural Land	Forest Area (Thousand ha)	Agricultural Land Area (Thousand ha)	The Density at 1000 ha	
					Forest	Agricultural Land
Nr. dogs	122	61	2606	13,597	46.8	4.5

**Table 4 vetsci-10-00188-t004:** The number of dogs caught, and the type of foods identified in each specimen.

Food Type	0	c	c,d	c,p	c,p,v	c,s	c,v	d	d,p	d,p,v	d,s	d,s,v	d,v	p	p,v	s	s,v	v	Total
Total	37	11	3	2	1	1	11	31	4	1	3	1	11	1	1	40	8	16	183

(where: 0—empty stomach, c—household organic matter, c,d—household matter+domestic meat, c,p—household matter+plastic, c,p,v—household matter+plastic+vegetables, c,s—domestic matter+wild animal, c,v—domestic matter+vegetables, d—domestic animal meat, d,p—domestic animal meat+plastic, d,p,v—domestic animal meat+plastic+vegetables, d,s—meat of domestic animal, wild, d,s,v—meat of domestic animal+wild animals, vegetables d,v—meat of domestic animal+vegetables, p—plastic, p,v—plastic+vegetables, s—meat of wild animal, s,v—wild animal meat+vegetables, vegetable matter).

**Table 5 vetsci-10-00188-t005:** Numerical distribution of dogs by type of food and by season over the total area analysed.

Food Type	0	c	c,d	c,p	c,p,v	c,s	c,v	d	d,p	d,p,v	d,s	d,s,v	d,v	p	p,v	s	s,v	v	Grand Total
W	15	4	-	-	-	-	4	5	1	-	1		6	-	-	16	-	9	61
Sp	6	3	-	2	-	1	4	12	1	-	1	1	2		-	11	3	2	49
S	9	-	-	-	-	-	-	5	2	-	1	--	-	1	-	6	-	1	25
A	7	4	3	-	1	-	3	9	-	1	-	-	3	-	1	7	5	4	48
Grand total	37	11	3	2	1	1	11	31	4	1	3	1	11	1	1	40	8	16	183

W—winter (21 December–20 March); Sp—spring (21 March–20 June); S—summer (21 June–20 September); A—autumn (21 September–20 December).

**Table 6 vetsci-10-00188-t006:** Percentage distribution of preferential food by season over the total area analysed.

Food	0	c	c,d	c,p	c,p,v	c,s	c,v	d	d,p	d,p,v	d,s	d,s,v	d,v	p	p,v	s	s,v	v	Total
W	24.6	6.6	-	-	-	-	6.6	8.2	1.6	-	1.6	-	9.8	-	-	26.2	-	14.8	100.0
Sp	12.2	6.1	-	4.1	-	2.0	8.2	24.5	2.0	-	2.0	2.0	4.1	-	-	22.4	6.1	4.1	100.0
S	36.0	-	-	-	-	-	-	20.0	8.0	-	4.0	-	-	4.0	-	24.0	0.0	4.0	100.0
A	14.6	8.3	6.3	-	2.1	-	6.3	18.8	-	2.1	-	-	6.3	-	2.1	14.6	10.4	8.3	100.0
Total	20.2	6.0	1.6	1.1	0.5	0.5	6.0	16.9	2.2	0.5	1.6	0.5	6.0	0.5	0.5	21.9	4.4	8.7	100.0

## Data Availability

On reasonable request, the derived data supporting the findings of this study are available from the corresponding authors.

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
