# Peer review of "Research on the Ethology and Diet of the Stray Dog Population in the Areas Bordering the Municipality of Suceava, Romania"

_vetsci, 2023, doi:10.3390/vetsci10030188_

Round 1

Reviewer 1 Report (Previous Reviewer 2)

I reviewed the previous version of the manuscript.  While the authors have addressed some of the concerns I raised (e.g. limitation of the research), I cannot see much improvement in the quality of the paper. The following issues still persist in the current version:

1) The research aim and questions are still not clearly stated.

2) A clear conceptual framework for the research is lacking. Currently, what is missing in the literature and what research gaps are to be filled are unclear.

3) Many of the dog photos presented in the paper is unnecessary. The authors should consider removing some of the photos from the paper.

4) The research design is not well explained in the paper. 

5) Can the findings of the research generalized?

6) Throughout the paper, there are a lot of short paragraphs. These paragraphs are not interlinked. I suggest the authors need to integrate the paragraphs and make sure there are logical transitions between paragraphs.

7) Besides, the paper has a lot of typos and grammatically errors. The authors should have the paper proofread by a professional English writer.

Author Response

Dear reviewer,

Thank you for your effort to review our paper and for your comments. As you suggested, I made changes:

- The purpose of the research is to highlight the behavior of stray dogs from the outskirts and the food consumed, in the area where we carry out our activity. In this area, in Romania, as well as in many countries of the world, stray dogs are a major problem, both from

ecological, as well as social and economic point of view;

- Such studies are very few, isolated and punctual. The reason is given by the sensitivity of the subject from the perspective of mass media and "animal lovers". The truth is that such works are "pioneering" and it takes a certain "courage" to be

published...

- It was mentioned that the results of ethological research can be generalized for similar social, economic, legislative and even political conditions.

- The results regarding food cannot be generalized. They depend on local conjunctural factors.

- Short paragraphs were made to be more accessible. As a rule, long sentences and large paragraphs result in a more cumbersome read;

- The translation was done by a British English teacher.

With thanks, and with the hope that no further changes will be needed,

Reviewer 2 Report (New Reviewer)

Research on the ethology and diet of the stray dog population in the areas bordering the Municipality of Suceava, Romania

The present article approaches the problem of stray dogs in a very complete way with interesting and valuable results. Currently, the population of stray dogs is increasing worldwide and, with it, the consequences of such events. Therefore, studying the causes but also the current view and possible alternatives to solve this issue is highly relevant. As a major comment, some of the sections need better organization for the reader to smoothly follow the Methods that the authors used and the results.

General comment regarding the citation style: Please, revise the Instructions for Author’s page and amend the citation style accordingly. It should be in numerical order inside brackets. In-text citation for Figures and Tables needs to be revised as well.

Lines 20-21: What is de meaning of “feral dogs betray their domestic origin”? It might be necessary to rephrase this sentence.

Introduction: In general, I would recommend changing the order of the paragraphs in this section. The authors could start by stating the dog-human bond (lines 25-29), the zootechnical uses of dogs (lines 44-45), and the worldwide dog population with the percentage that corresponds to stray dogs (lines 41-43).

 Then, the next lines could be about why although such a strong bond has been established since ancient times, recently the number of stray dogs is increasing due to the reason that is listed in line 31. After that, the main issues associated with stray dogs could be mentioned (from line 46 to the end of the Introduction).

Lines 25-29: The first paragraph of the Introduction needs to improve. To give relevance to the human-animal bond that dogs and humans share since ancient times, it might be appropriate to include the age at which the discoveries in the Egyptian cemetery remount and highlight the bond that was formed between the two species. The pathological lesions and disease need to be mentioned somehow to understand the relation of lesions in dogs to their dependence on human care.

Lines 30-32: Apart from the reasons that are already listed in these lines, animal abandonment is one of the reasons why the number of stray dogs keeps increasing over the world. Please, revise this article:  https://doi.org/10.14202/vetworld.2021.2371-2379 

Lines 41-42: Could you use a more updated reference than the one from 1990 of the worldwide dog population, please?

Lines 48-52: Could you please write a short paragraph that goes a little deeper into the zoonoses? I consider that it could be an important point to deepen due to the importance that these diseases have within society.

Lines 68-69: Could you please attach a small statement that talks about the pollution that the feces and the dogs themselves when spreading garbage could generate, as well as that they could also cause traffic accidents?

Line 121: Please, state the aim of the article before section 2.

Line 150: It is stated that “observations on the itinerary were made” but there is no previous description of what the itinerary is. I would also recommend adding if the observation were made at certain hours of the day (due to the activity pattern of the dogs in the wild).

Lines 157-159: Consider moving this paragraph to the introduction, so the definition of “stray dog” is stated since the beginning.

Lines 186-190: Could the authors provide a reference for the legislation where shooting is the recommended method to capture stray dogs?

Line 191: Add a reference for the dissection technique used in the present article.

Lines 219-221: In the statistical analysis there needs to be a clear description of the type of analysis performed for every assessed variable.

Line 496: Before mentioning the plant species that can be found in the studied area, I recommend starting the discussion by making a summary of the main findings. For example, how many stray dogs were found, the main issues regarding stray dogs (e.g., hunting livestock), the kind of food that stray dogs consume the most, etc?

Also, as a general comment about the discussion, I find the number of references in this section very low to be a discussion.

Line 502: Revise the font size of “Campanula persicifolia”

Lines 496 and 518: Both sentences begin with the same words. Please, rephrase.

Lines 552-559: Instead of just mentioning the results of the present study, it might be more interesting to compare the feeding behavior of stray dogs with companion dogs and then to wild carnivores, from a behavioral, anatomical, and physiological perspective. In this way, the reader can understand the impact of the domestication of dogs and the adaptation of the animal to the wild.

Lines 580-588: Could it be that most of the stray dogs were captured in winter, in addition to the causes you mention, since they may require more food (predation) more frequently because they have higher energy requirements to maintain their body temperature? I think that point would still be good to mention as part of the discussion.

Line 603: Before the conclusion, the authors could mention some limitations of the study and some recommendations regarding stray dogs. Also, one issue that requires further discussion is the location of stray dogs near urbanized areas (to obtain food) and the public health issues that this might cause not only to humans but to other companion animals, as well as the economic losses that dogs’ attack cause on livestock.

Author Response

Dear reviewer,

Thank you for your effort to review our paper and for your comments. As you suggested, I made changes:

20-21: Redone.

25-29, 30-32, 41-43, 44-45: Redone.

41-42: Reference.

48-52: Extended.

121: Done.

150: Redone.

157-159: Moved to introduction;

186-190: Explained.

2019-221: Redone.

496: We believe that the biodiversity of the area should be briefly presented at the beginning of the discussions. The rest of the discussions you request are presented in stages, in the order in which the results are presented.

502: Redone.

496, 518: Redone.

580-588: Redone.

603: Redone with explanations.

With thanks, and with the hope that no further changes will be needed,

This manuscript is a resubmission of an earlier submission. The following is a list of the peer review reports and author responses from that submission.

Round 1

Reviewer 1 Report

This report treats a subject likely to be of interest to a wide readership given the importance of the domestic dog as a companion animal and the still rather sparse information on the basic ethology and ecology of free-raging dogs. Nevertheless, although I appreciate the difficulty of conducting such studies, I have several concerns which I believe need to be addressed before this submission can be considered for publication.      

Major concern

Much of the information is anecdotal, consisting in a series of subjective descriptions and assertions without sufficient quantitative evidence. Lacking is detailed information of the data base on which such assertions are based; sample sizes, including number of reliably identified individuals, how they were identified, number of observations of each individual, verified by how many independent observers. Also lacking are definitions of stray, feral and wild dogs and how individuals were identified as belonging to these categories, and it is not even clear, for example Table 7, what the authors considered to be a small, middle or large dog. Although the sample of 183 specimens “captured”, presumably shot (this needs to be clarified), is considerable, the method of identifying stomach/bowel contents (not clear) is not adequately described; “analyzed organoleptically” is not sufficient – visually, by smell, presumably not by taste! Also, without a test of independent inter-rater reliability of these evaluations it is difficult to have confidence in the ability of the investigators to distinguish, for example, between the remains of domestic and wild animals.

I recommend that the authors eliminate most of the dog photos which contribute no scientific information, and use the space saved to give a more adequate account of the methodology. Without this it is not possible to judge the scientific value of the report.            

And a couple of minor points

- Avoid the word “preference”. Based on the data presented, it is not known if the stomach/bowel contents of the dogs sampled represented their food preferences or simply what had been (opportunistically?) available to them on the day of sampling. On another day the results might have been different. Just describe the findings without interpreting them as evidence of preferences, which to decide would normally involve some kind of choice tests.

- I dispute the statement (L. 27) that cats have become dependent on human presence. Although cats can certainly take advantage of human presence in various ways, they are also capable (unlike domestic dogs) of leading lives completely independent of human contact and of forming reproductively viable populations, sometimes to plague proportions such as on many islands and in outback Australia. So simply eliminate the cats. They contribute nothing to this study.

Reviewer 2 Report

The research aim and questions are not clear. After reading the paper twice, I still could not quite catch what the study really wants to aim to achieve.

I suggest the authors to provide a clear conceptual framework for the research. Currently, what is missing in the literature and what research gaps are to be filled are unclear.

The research design is not well explained in the paper. 

Can the findings of the research generalized?

What are the limitations of the research? How do these limitations affect the interpretations of the research findings?

Throughout the paper, there are a lot of short paragraphs. These paragraphs are not interlinked. I suggest the authors need to integrate the paragraphs and make sure there are logical transitions between paragraphs.

Besides, the paper has a lot of typos and grammatically errors. The authors should have the paper proofread by a professional English writer.